# A Molecular Dynamics Approach to Explore the Intramolecular Signal Transduction of PPAR-α

**DOI:** 10.3390/ijms20071666

**Published:** 2019-04-03

**Authors:** Shaherin Basith, Balachandran Manavalan, Tae Hwan Shin, Gwang Lee

**Affiliations:** Department of Physiology, Ajou University School of Medicine, Suwon 443-749, Korea; shaherinb@aumc.ac.kr (S.B.); bala@ajou.ac.kr (B.M.); catholicon@ajou.ac.kr (T.H.S.)

**Keywords:** PPAR-α, molecular dynamics, apo, agonist, antagonist, hydrogen bond, network theory, betweenness centrality

## Abstract

Dynamics and functions of the peroxisome proliferator-activated receptor (PPAR)-α are modulated by the types of ligands that bind to the orthosteric sites. While several X-ray crystal structures of PPAR-α have been determined in their agonist-bound forms, detailed structural information in their apo and antagonist-bound states are still lacking. To address these limitations, we apply unbiased molecular dynamics simulations to three different PPAR-α systems to determine their modulatory mechanisms. Herein, we performed hydrogen bond and essential dynamics analyses to identify the important residues involved in polar interactions and conformational structural variations, respectively. Furthermore, betweenness centrality network analysis was carried out to identify key residues for intramolecular signaling. The differences observed in the intramolecular signal flow between apo, agonist- and antagonist-bound forms of PPAR-α will be useful for calculating maps of information flow and identifying key residues crucial for signal transductions. The predictions derived from our analysis will be of great help to medicinal chemists in the design of effective PPAR-α modulators and additionally in understanding their regulation and signal transductions.

## 1. Introduction

Peroxisome proliferator-activated receptors (PPARs) are members of the nuclear hormone receptor superfamily that bind to varied signals and transduce them into an ordered set of cellular responses at the level of gene transcription [1,2,3]. So far, three mammalian PPAR subtypes, namely PPARα, PPARβ/δ, and PPARγ have been identified, which are encoded by distinct genes [1]. PPARs are activated by several endogenous ligands, including cholesterol metabolites, retinoids, saturated and unsaturated fatty acids, steroids, and pharmacological compounds [4,5]. All PPARs share a common structural architecture consisting of a DNA-binding domain at the N-terminus and a ligand-binding domain (LBD) at the C-terminus [6]. Upon ligand binding, PPARs are translocated to the nucleus and heterodimerize with the retinoid X receptor. This complex then binds to coactivators for interacting with specific DNA motifs known as peroxisome proliferator-response elements and regulates the expression of specific target genes [7]. Although PPAR subtypes share uniqueness in their tissue distribution, they show distinct regulatory and modulatory activities.

PPAR-α is the first PPAR subtype to be cloned from mouse-liver complementary DNA-library as a nuclear receptor that causes proliferation of peroxisomes [8]. PPAR-α is a transcription factor that is activated by several natural and synthetic ligands. This receptor subtype acts as a key regulator of energy metabolism and mitochondrial and peroxisomal function via target gene regulation [9,10]. Furthermore, it is involved in the activation of fatty acid β-oxidation and ketogenesis, and simultaneous inhibition of glycolysis and fatty acid synthesis [11,12,13]. Considering their wide range of actions on lipid and glucose metabolism, control of inflammatory processes, and vascular integrity maintenance, PPAR-α and their modulators are recommended for the treatment of metabolic diseases, such as atherosclerosis, dyslipidemia, and hyperglycemia [6].

Until now, only few crystal structures of PPAR-α LBD with their respective co-crystallized ligands have been solved. Previous studies suggest that agonists form polar interactions with S280, Y314, A333, H440, and Y464, whereas antagonists form a polar interaction with Y314, which are responsible for agonist and antagonist recognition, respectively [14,15]. As crystal structures represent only a snapshot of the system that has been captured in a particular state, information about other key residues located in the binding pocket that could participate in covalent or non-covalent interactions with ligands and could contribute to ligand recognition is still lacking. Therefore, molecular dynamics (MD) simulations that capture the receptor motions at an atomic scale could help in connecting those dots [16].

Previously, Liu et al. performed 50 ns MD simulation of ligand-bound PPAR-α complexes and suggested a few key residues involved in ligand-recognition using H-bond and energy decomposition analyses [17]. However, intrinsic structural information on apo and antagonist-bound forms of PPAR-α LBD remains inadequate. To the best of our knowledge, no apo-form PPAR-α LBD structure model has been proposed and no prolonged simulation studies for the three different states of PPAR-α LBDs (apo, agonist-, and antagonist-bound forms of PPAR-α) have been performed so far. In this study, we are the first to explore the conformational dynamics for three different states of PPAR-α LBDs using conventional MD simulations. We performed 300 ns all-atom MD simulations for each PPAR-α LBD in apo, agonist- (13M), and antagonist-bound (471) forms and analyzed their trajectories using several structural and network analyses. The structural insights gained from this study opens the gateway for large-scale virtual screening and could assist medicinal chemists in the design of novel and effective PPAR-α ligands. This approach could also be extended to other PPAR subtypes and nuclear receptors.

## 2. Results and Discussion

### 2.1. Structural Stability of PPAR-α Complexes During Simulations

Initially, we prepared three different states of PPAR-α systems and subjected each one of them to 300 ns MD simulations. To assess the stability of the complexes, we calculated the root mean square deviation (RMSD) of the protein backbone atoms with respect to the equilibrated structure as a function of the simulation time. As expected, the apo form showed high structural deviations ranging from 2.5–6 Å (Figure 1A), thus suggesting that the unbound form of PPAR-α may be highly unstable in physiological conditions. Whereas, the ligand-bound complexes exhibited an overall deviation of less than 2 Å, signifying their greater structural stability in agonist- or antagonist-bound states. It was observed that both ligand-bound complexes reached structural stability after 220 ns simulation time (Figure 1A) and the structural deviation of the agonist-bound was lower than the antagonist-bound complex. Overall, all three systems deviated from the equilibrated structures in the range of 1.2–6 Å during production run. Similarly, ligand RMSD (Figure 1B) was calculated for 13M (agonist) and 471 (antagonist). 13M RMSD ranged from 0.5–2.25 Å, whereas 471 RMSD ranged from 1–2.5 Å. Both ligands showed stability after 200 ns production run.

The root mean square fluctuations (RMSFs) of Cα atoms were calculated for each residue in all systems. Predictably, the residues in the terminal regions (N- and C-terminal ends) showed high fluctuations up to 9 Å, especially in the apo form (Figure 1C), since they tend to be more exposed on the surface with greater mobility. Whereas, the residues involved in ligand binding were relatively stable and maintained their fluctuations within 1 Å (shown as black arrow in Figure 1C). For clarity, the residues that were highly fluctuating (> 3 Å) during the production runs are displayed in all three systems in Figure 1C. Besides RMSD, solvent accessible surface area (SASA) is also employed to assess the stability of the systems during simulation [18]. We calculated the total SASA for the backbone atoms of all three systems (Figure 1D). The average SASA for apo, agonist-, and antagonist-bound systems were 150.448 nm^2^, 145.592 nm^2^, and 160.143 nm^2^, respectively. In Figure 1D, the agonist-bound system showed lower SASA compared to the other two systems (apo and antagonist-bound forms), thus indicating its higher thermodynamic stability. Furthermore, the SASA of the antagonist-bound form is highly stable and does not change during simulation. Interestingly, the total SASA of apo form lies between agonist- and antagonist-bound forms. Overall, our trajectory analysis showed the stable nature of the systems and were thus utilized for further analysis.

### 2.2. Hydrogen Bond Analysis

Hydrogen bonds (H-bonds) are facilitators in protein-ligand systems to stabilize the ligand in the binding pocket [19]. Practically, it is not possible to check H-bond stability by inspecting the crystal, nuclear magnetic resonance, or electron microscopy-solved structures. Therefore, it is essential to check the stability of the H-bonds using MD simulations. To identify the polar interactions in PPAR-α ligand-bound states, we calculated and analyzed the H-bonds using a gmx hbond tool that determines the presence of H-bonds based on a cutoff distance of 3.5 Å and a cut-off angle of 30° [20]. Initially, we calculated the number of H-bonds between the ligand and protein (Figure 2A) and identified that they varied throughout the simulation. Notably, the maximum number of H-bonds in agonist- and antagonist-bound states is 4 and 6, respectively. Moreover, approximately two H-bonds were found to be sustained in the ligand-bound complexes throughout the production runs.

To identify the residues involved in H-bond formation for agonist or antagonist recognition, H-bond occupancies were calculated. H-bond occupancy refers to the fraction of time in which the ligand atom forms H-bonds with any of the interacting residue in the protein. In Figure 2B, it was observed that Q277 and A333 were mainly involved in H-bond formation for the PPAR-α/13M system, which agrees with a previous report [14]. Other minor residues that are involved in the polar interactions for agonist recognition are T279, S280, Y314, I354, K358, and H440. Similarly, the major residues involved in H-bond formation for antagonist recognition are S280, Y314, and H440. Previously, it was shown that Y314 was involved in polar interaction in the PPAR-α/471 system, which agrees with our analysis [15]. Other minor residues that are involved in H-bond formation for antagonist recognition are Q277 and T279. Overall, our H-bond analysis signifies the importance of polar interactions in the selective recognition of agonist and antagonist molecules.

### 2.3. Essential Dynamics Analysis

Principal component analysis (PCA) determines the dominant motions accountable for the functional behaviors of proteins. The two-dimensional (2D) projection of PC1 and PC2 (between the largest two eigenvalues) in all PPAR-α systems is plotted in Figure 3A. In Figure 3A, each dot represents the conformation of PPAR-α in the apo and ligand-bound forms over a trajectory of 300 ns. For larger conformational changes, the spreading of distribution will be more in the conformational space. It is observed that the apo form distributions are widely scattered due to the absence of a ligand, whereas the conformational space of PPAR-α in the ligand-bound forms are limited. The plotted eigenvalues (nm^2^) against the eigenvector index showed stabilization in the first seven eigenvectors (Figure 3B). Additionally, the cumulative percentages of variance in motion obtained from the first 20 principal components were 88%, 22%, and 35% for apo, agonist-, and antagonist-bound systems, respectively (Figure 3B). All these principal component analyses suggest that the apo form of PPAR-α shows a large distribution of structural conformations in comparison to the ligand-bound forms. In ligand-bound forms, the agonist-bound system shows lesser distribution of structural conformations in comparison to the antagonist-bound form, thus indicating that PPAR-α is highly stable upon agonist binding.

The global minimum energy conformation of the protein-ligand complex can be obtained through free energy landscape (FEL) analysis. The FEL contour maps for all three PPAR-α systems were derived from PC1 and PC2 (as mentioned in the methodology), which are depicted in Figure 4 and Figure 5.

The 2D-FEL contour maps for apo and antagonist-bound forms showed two minimal energy clusters and large structural distributions (Figure 4A,B and Figure 5D,F). However, in the agonist-bound form, there is only a single conformation cluster that indicates strong and stable interaction of agonist with PPAR-α, which agrees with our structural stability analysis. Furthermore, the least energy or the most representative conformation from the least minimal energy regions in ligand-bound forms were extracted for showing the stable interactions between the protein and ligand. The FEL analysis of the agonist- and antagonist-bound forms revealed that the access to the lowest energy conformer was at the 64,500 ps and 176,300 ps snapshots, respectively, which were extracted from the most populated free energy minimum cluster. In agonist-bound conformation, the agonist 13M forms four stable H-bonds with Q277, A333, and Y464 residues (Figure 5B), which is in agreement with a previous study [14] and supported well by our H-bond analysis. Likewise, antagonist 471 forms five stable H-bonds with Q277, S280, Y314, and H440 residues (Figure 5E), which agrees well with a previous report [15] and our H-bond analysis.

### 2.4. Generation of PPAR-α Minimal Energy Structures

Selection of a representative structure from the MD trajectory frames for analysis is one of the biggest challenges in the utilization of large structural ensembles. The choice of the selection criteria may vary depending on the goal of our subsequent analysis. The representative structure could be selected based on the RMSD, total conformational energy, or certain known conformational changes, which may be crucial for protein function. In our protocol, the trajectories of the MD production run for each system were monitored based on the total conformational energy (Figure 6A) and RMSD relative to the lowest conformational energy structure (Figure 6B). The lowest conformational energy structures for apo, agonist-, and antagonist-bound forms were obtained by plotting the total conformational energy of the system relative to the RMSD of the protein backbone atoms (calculated with respect to the equilibrated structure), which are highlighted as red circles in Figure 6C. The top 20 structures with lowest energies were clustered for all three PPAR-α systems. The top minimal energy conformation structures obtained from the conformational ensembles were selected as the representative structures for the apo, agonist-, and antagonist-bound forms of PPAR-α, which were subsequently subjected to network analysis (Figure 6D).

### 2.5. Network Centrality Analysis

Measuring centrality is a popular concept in social network analysis, which could increase the applicability of selecting critical nodes in the network [21,22,23]. Centrality calculations have been successfully applied in identifying functionally important residues, such as those present in the active site or co-factor binding site of a protein, protein allosteric communications, metabolic networks, and disease networks [24,25,26]. The three most widely applied centrality measures are betweenness (*C_B_*), closeness (*C_C_*), and degree (*C_D_*) [21,22]. To identify the crucial residues responsible for agonist or antagonist signal flow, we used the minimum energy structures of apo, agonist-, and antagonist-bound states of PPAR-α, constructed the residue interaction networks (as mentioned in methods section), and computed the centrality measures (Figure 7). Notably, the overall Pearson correlation coefficient between three different centralities lies in the range of 0.66–0.72, hence a residue with high *C_C_* or *C_D_* does not necessarily possess a high *C_B_* value. Previous studies have shown that a node with high *C_B_* values correlates well with the functional residues that mediate allosteric signals, protein-protein interactions, etc., indicating the importance of *C_B_* over other two centralities [27,28]. Henceforth, only *C_B_* was calculated to measure the importance of each residue for the signal flow in three different states of PPAR-α [23].

We mapped the residues with *C_B_* ≥ 0.05, which constitute the top 10% of the *C_B_* distribution, onto the minimized apo structure of PPAR-α (Figure 7D). Ten residues were identified using *C_B_* ≥ 0.05 and a few residues among them are deemed important for agonist and antagonist signaling (Table 1), which are in line with previous studies [14,15]. Remarkably, these residues are primarily distributed contiguously in the active site and C-terminal regions, and thus could be deemed crucial for modulating the signal flow.

Similarly, we also calculated the differences in the *C_B_* between apo and agonist-bound forms (Figure 8A), and between apo and antagonist-bound forms (Figure 8B). The contribution of residues satisfying the condition |*C_B_^Apo^* − *C_B_^Ago^*| ≥ 0.02 and |*C_B_^Apo^* − *C_B_^Antag^*| ≥ 0.02 are mainly located in the orthosteric site and C-terminal ends also satisfying *C_B_* ≥ 0.05 (Table 2).

Thirteen and fifteen residues were identified using |*C_B_^Apo^* − *C_B_^Ago^*| ≥ 0.02 and |*C_B_^Apo^* − *C_B_^Antag^*| ≥ 0.02, respectively. Few among those residues are in line with previous studies for agonist or antagonist recognition (see Table 2 for details) [14,15]. For legibility purpose, the residues that satisfy the condition |*C_B_^Apo^* − *C_B_^Ago^*| ≥ 0.03 and |*C_B_^Apo^* − *C_B_^Antag^*| ≥ 0.03 are only depicted in the minimized agonist- (Figure 8C) and antagonist-bound (Figure 8D) structures of PPAR-α.

## 3. Materials and Methods

### 3.1. Computing and Data Analysis

All computing was carried out on a Linux (CentOS release 7.6.1810) workstation. The trajectories were analyzed using GROMACS v5.1.4. built-in modules [29,30]. FEL contour maps were plotted using a trial version of Mathematica 11.3. Network analysis was performed using our in-house script. The graphical images and plots were produced using PyMOL v2.3.0a0. [31] and Matplotlib v3.0.3. [32], respectively.

### 3.2. Preparation of the Apo, Agonist-, and Antagonist-Bound Structures

Prior to MD simulations, both ligand and receptor structures should be prepared. High resolution crystal structures of PPAR-α/13M (PDB code: 3VI8) [14] and PPAR-α/471 (PDB code: 1KKQ) [15] complexes were obtained from the Research Collaboratory for Structural Bioinformatics (RCSB) Protein Data Bank (PDB). Crystal water molecules within 4 Å of ligands were retained. For PPAR-α/471, only chain A complexed with a ligand molecule was considered for analysis, since this work is focused on exploring the interactions of PPAR-α with ligands. Although several crystal structures of PPAR-α are determined in their agonist- and antagonist-bound forms, structural data for apo form is still lacking. The apo form of PPAR-α was obtained by minimizing the inactive PPAR-α structure (PDB code: 1KKQ), after the removal of the bound antagonist. The agonist-bound structure of PPAR-α was modeled using ModBase [33]. We used the structure with PDB ID 2P54 [34] as a template for the agonist-bound form in order to generate models for the loop regions that were not determined in 3VI8. Furthermore, ligand molecules were prepared using Open Babel software v2.3.2 [35].

### 3.3. Molecular Dynamics (MD) Simulations

The prepared systems were subjected to MD simulations using GROMACS v5.1.4. with CHARMM36-nov2018 force field. The topology and parameter files for the ligands were generated using CGenFF program v2.2.0 [36]. The systems were solvated with 18,852 (apo), 13,221 (agonist-bound), and 12,915 (antagonist-bound) water molecules using TIP3P water model and were subsequently neutralized by replacing the solvent molecules with required counter ions. All the systems were minimized using the steepest descent algorithm, and the tolerance force for convergence was set to <1000.0 kJ (mol nm)^−1^. Subsequently, the systems were subjected to first equilibration phase with the constant volume (NVT) ensemble for 10 ns each using leap-frog integrator to attain the desired temperature (300 K). Furthermore, the systems were subjected to a second equilibration phase with the constant pressure (NPT) ensemble for 10 ns each using a Parrinello-Rahman barostat to attain 1 bar pressure. Nonbonded interactions were smoothly switched off between 10 and 12 Å. To handle electrostatic interactions, the particle mesh Ewald algorithm was employed with a cubic interpolation order of 4 and a grid spacing of 0.16 nm. Upon the completion of equilibration, the systems were finally subjected to 300 ns production run with NPT ensemble. We used the integration time step of 2 fs, and for the analysis we saved and sampled the simulated trajectories for every 100 ps.

### 3.4. Principal Component Analysis (PCA) and Free Energy Landscape (FEL)

PCA could be utilized to identify the high-amplitude concerted motions along the MD trajectory [37,38,39]. Firstly, covariance matrices for three different PPAR-α systems were constructed from protein backbone atomic fluctuations after the removal of rotational and translational motions. From diagonalization of the covariance matrix, the sum of eigenvalues for apo, agonist-, and antagonist-bound forms were calculated to be 38.7978 nm^2^, 11.3491 nm^2^, and 16.8518 nm^2^, respectively. GROMACS tool gmx covar was utilized for calculating and diagonalizing the covariance matrix of backbone atoms and obtaining the eigenvalues and eigenvectors [40]. The gmx anaeig was utilized for analyzing the eigenvectors [41]. The first 20 projection eigenvectors of the proteins were extracted from simulated PPAR-α complexes and analyzed for their cosine content in which the first two eigenvectors, PC1 and PC2 having a cosine content less than or equal to 0.2, were used to define the FEL [42]. The FEL of the three systems were mapped to obtain minimum energy configurations using the gmx sham tool of the GROMACS package from the two PCs based on cosine content analysis [43]. The contour maps of the FEL were generated using a trial version of Mathematica 11.3.

### 3.5. Residue Interaction Network Construction

A weighted residue–residue interaction network was constructed, where each amino acid and number of H-bonds between two residues respectively represent the node and weight. Two coarse-grained centers per residue was considered by taking into account the effect of side chain, i.e., Cα backbone and heavy atoms far away from Cα for the side chains. Basically, we considered the cases of backbone-sidechain, side chain-side chain, and backbone-backbone contacts. In our network model, a contact was established if any one of the above defined atoms between two residues was less than 7 Å.

### 3.6. Network Centralities

Using residue interaction network, we computed three types of centralities, namely *C_D_*, *C_C_*, and *C_B_*, the definitions of which are as follows: *C_D_(k)* measures the total number of edges linked to a node *k*, which is identical to the number of contacts with its neighboring residues.
(1)$$C_{D}\left(k\right)=\deg\left(k\right)$$
*C_C_(k)* of a node *k* is the reciprocal of the average shortest path distance from all other nodes to the node *k*, basically it measures how fast a signal could be transmitted to other nodes from the node *k*.
(2)$$C_{c}\left(k\right)={\left(\overset{N}{\underset{i=1}{\sum }}d\left(i,\;k\right)/\left(N-1\right)\right)}^{-1}$$
where d(*i*, *k*) is the minimum number of edges that bridge the node *i* and *k*. Dijkstra’s algorithm was employed to compute d(*i*, *k*).

*C_B_(k)* is used to identify the central node, which is crucial for signal transduction within the protein. Brandes algorithm [44] was employed to compute the *C_B_*.
(3)$$C_{B}=\;\frac{2}{\left(N-2\right)\left(N-1\right)}\overset{N-1}{\underset{s=1}{\sum }}\overset{N}{\underset{t=s+1}{\sum }}\frac{\sigma_{st}\left(k\right)}{\sigma_{st}}$$
where $\sigma_{st}$ is the number of shortest paths connecting the nodes *s* and *t*, and $\sigma_{st}\left(k\right)$ is the number of shortest paths connecting the nodes *s* and *t* through the node *v*; $\frac{2}{\left(N-2\right)\left(N-1\right)}$ is the normalization constant.

## 4. Conclusions

Although the significance of PPAR-α has been widely acknowledged, the conformational changes observed in three different states is still lacking. Therefore, we utilized unbiased MD simulations for three PPAR-α systems and performed the analysis on each 300 ns MD trajectory. Structural stability analysis showed a more stable nature of agonist-bound form of PPAR-α when compared to the other two systems. H-bond analysis showed the significance of polar interactions in the agonist and antagonist recognitions and identified both major and minor residues involvement in H-bonds, which are in line with previous studies. Subsequently, PCA and FEL analyses showed the scattered distribution of the apo system when compared to the ligand-bound systems of PPAR-α. Additionally, the lowest energy conformers for agonist- and antagonist-bound forms obtained from the FEL analysis further strengthen our H-bond analysis. The generated minimal energy structures for the three systems were subjected to network analysis, where several crucial residues important for signal flow were identified. Notably, the predictions from the *C_B_*-based network analysis of protein structures should be of great use not only to this PPAR-α system but also to illuminate the origin of subtype selectivity and their modulation mechanisms. Overall, our approach is not only applicable for studying the conformational states of protein-ligand systems and mapping their associated signal flow but could also be extended to understand the communications in protein-protein, protein-RNA, or protein-DNA complexes as well.

## Figures and Tables

**Figure 1 ijms-20-01666-f001:**
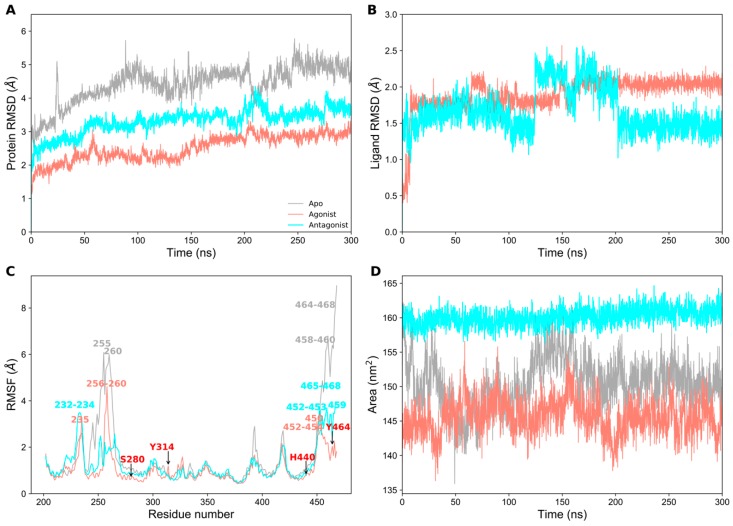
Analysis of the 300 ns molecular dynamics (MD) simulation productive phases of apo, agonist-, and antagonist-bound forms of PPAR-α. (**A**) Root mean square deviation (RMSD) of the protein backbone atoms with respect to the equilibrated structure; (**B**) RMSD of the agonist (13M) and antagonist (471) molecules with respect to the minimized structures; (**C**) Root mean square fluctuations (RMSF) of apo, agonist-, and antagonist-bound forms of PPAR-α and the residues with high fluctuations are shown. The residues important for ligand binding are labeled in red. (**D**) Solvent accessible surface area of apo, agonist-, and antagonist-bound forms of PPAR-α. The data for apo, agonist-, and antagonist-bound forms of PPAR-α are shown in gray, salmon, and cyan, respectively.

**Figure 2 ijms-20-01666-f002:**
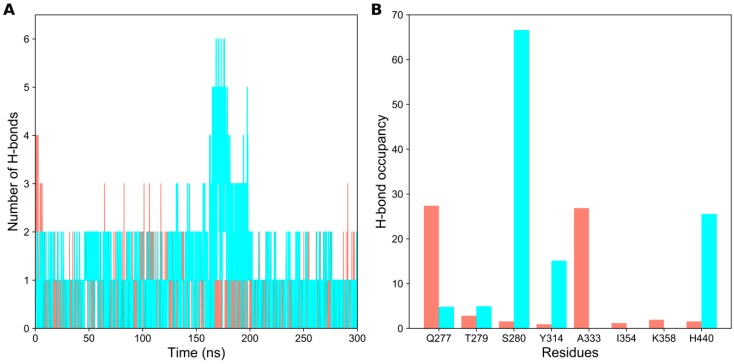
Hydrogen bond (H-bond) formation of agonist- and antagonist-bound forms of PPAR-α during 300 ns MD simulations. (**A**) Number of H-bonds formed between protein-ligand complexes with respect to the production period; and (**B**) H-bond occupancy of each interacting residue in their relevant complexes throughout the simulation. The data for agonist- and antagonist-bound forms of PPAR-α are shown in salmon and cyan colors, respectively.

**Figure 3 ijms-20-01666-f003:**
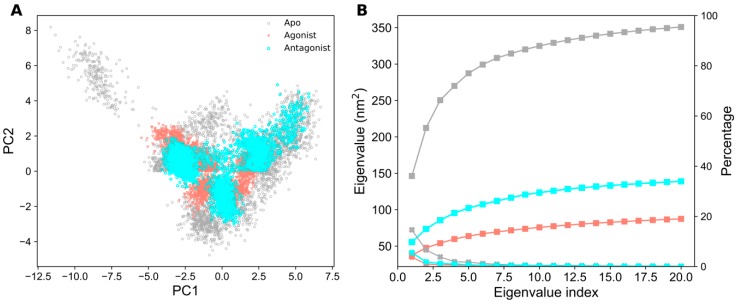
Projection of the different structural conformations onto PC1 and PC2 for three PPAR-α systems. (**A**) Two-dimensional (2D) projection of the selected eigenvectors throughout 300 ns MD simulation. Data for apo, agonist-, and antagonist-bound forms of PPAR-α are represented in gray, salmon, and cyan colors, respectively. (**B**) The best principal components of eigenvalues that correspond to the first 20 eigenvectors are represented by the asterisk line point and their associated cumulative fluctuations are represented by the square line point.

**Figure 4 ijms-20-01666-f004:**
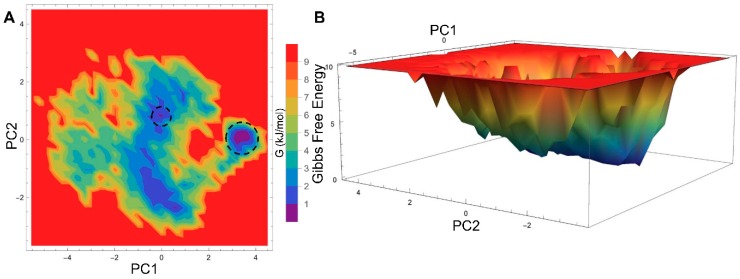
2D free energy landscape (FEL) of PPAR-α apo form (**A**,**B**) displayed as a function of two principal components (PC1 and PC2), whose cosine content was less than 0.2. The minimal energy clusters are highlighted using black dotted circles.

**Figure 5 ijms-20-01666-f005:**
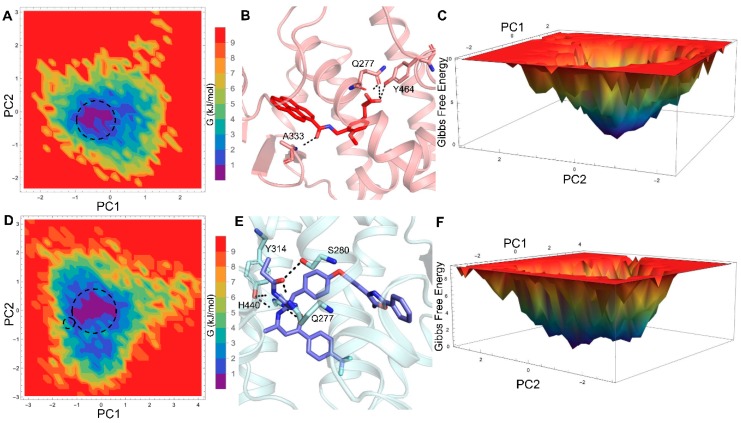
2D-FEL of agonist- (**A**,**C**) and antagonist-bound (**D**,**F**) forms of PPAR-α displayed as a function of two principal components, whose cosine contents were less than 0.2. The representative structures of agonist- (**B**) and antagonist-bound (**E**) forms with minimal energy are zoomed out to show the residues critical for ligand binding. The residues important for ligand recognition are shown as red and slate blue colored sticks and the H-bonds are displayed as black dashed lines. The minimal energy clusters are highlighted using black dotted circles.

**Figure 6 ijms-20-01666-f006:**
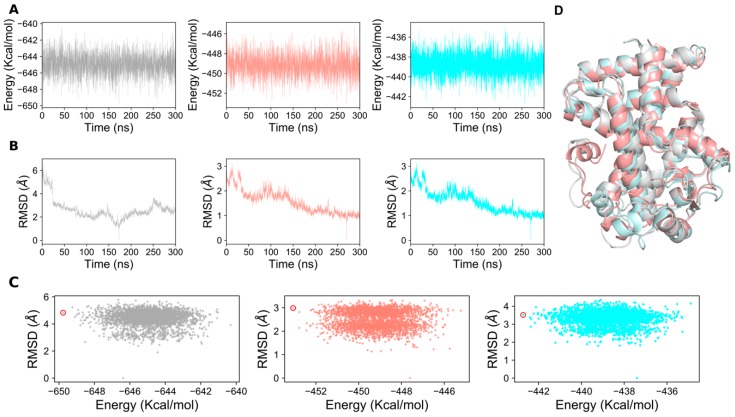
Structure and dynamics of PPAR-α complexes. (**A**) Time-dependent total conformational energy of apo, agonist-, and antagonist-bound forms; (**B**) RMSD in reference to the minimum energy structure from the MD trajectories of the apo, agonist-, and antagonist-bound forms; (**C**) RMSD plot with respect to the total conformational energies of apo, agonist- and antagonist-bound complexes. For clarity purpose, minimal energy structures of PPAR-α complexes are illustrated in red circle. (**D**) The minimum energy structures of apo, agonist, and antagonist-bound forms are overlaid to show the significant structural difference between the three forms. Gray represents apo-form, salmon represents agonist-bound form, and cyan represents antagonist-bound form of PPAR-α.

**Figure 7 ijms-20-01666-f007:**
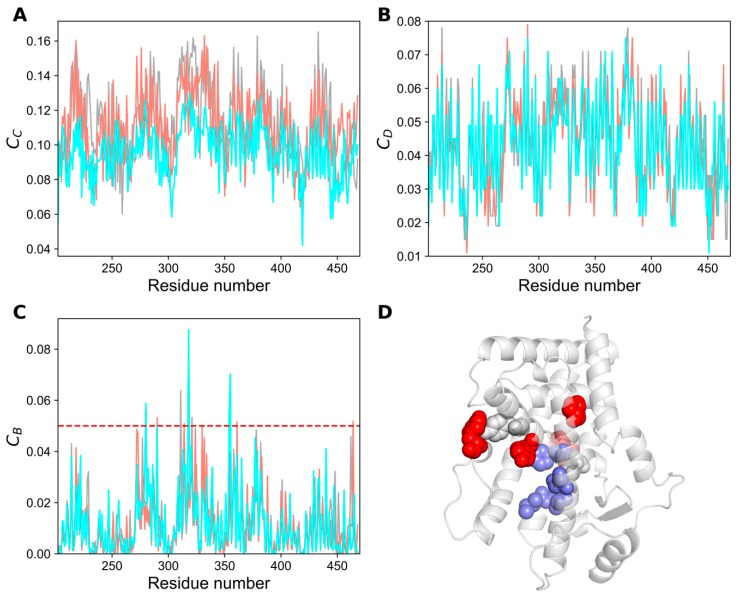
Network centrality analysis for three different systems of PPAR-α. (**A**) Closeness (*C_c_*), (**B**) degree (*C_D_*), and (**C**) betweenness (*C_B_*) centralities for each residue of the PPAR-α complexes are shown. (**D**) Residues with high *C_B_* (≥ 0.05) are depicted as spheres on the representative apo structure of PPAR-α. Gray, red, and slate blue spheres represent the residues with high *C_B_* values in apo, agonist-, and antagonist-bound forms of PPAR-α, respectively.

**Figure 8 ijms-20-01666-f008:**
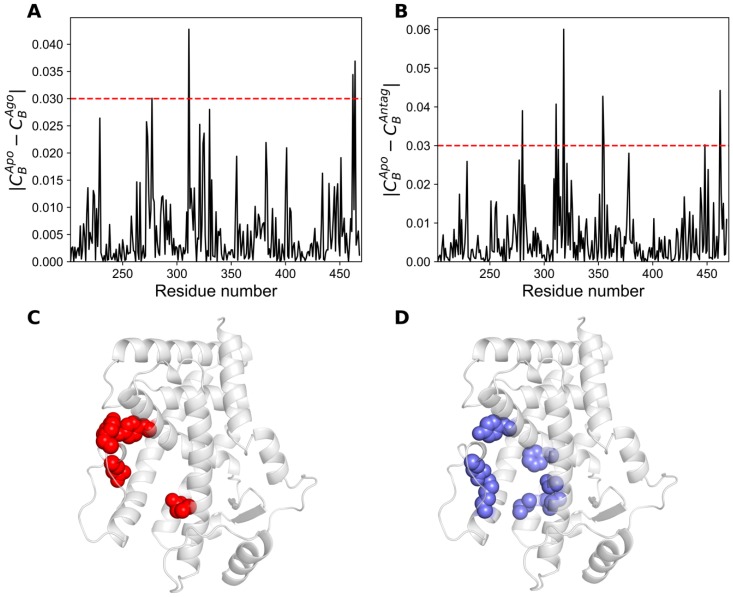
Difference of *C_B_* values calculated for (**A**) apo and agonist-bound structures. (**C**) Residues with |*C_B_^Apo^* − *C_B_^Ago^*| ≥ 0.03 depicted on the apo minimized structure as red spheres, contributed mainly from the orthosteric pocket and C-terminal region; (**B**) difference of *C_B_* values calculated for apo and antagonist-bound structures; (**D**) similarly, residues with |*C_B_^Apo^* − *C_B_^Antag^*| ≥ 0.03 depicted on the apo minimized structure as slate blue spheres, contributed primarily from the ligand binding pocket and C-terminal region.

**Table 1 ijms-20-01666-t001:** List of key residues with betweenness centrality values (*C_B_*) ≥ 0.05.

*C_B_^Apo^* ≥ 0.05	*C_B_^Ago^* ≥ 0.05	*C_B_^Antag^* ≥ 0.05
Y311, L321	F290, **Y314**, F361, **Y464**	**S280**, F318, I354, M355

Bold residues represent the key residues identified from previous studies.

**Table 2 ijms-20-01666-t002:** List of key residues satisfying the condition |*C_B_^Apo^* − *C_B_^Ago^*| ≥ 0.02 and |*C_B_^Apo^* − *C_B_^Antag^*| ≥ 0.02.

|*C_B_^Apo^* − *C_B_^Ago^*| ≥ 0.02	|*C_B_^Apo^* − *C_B_^Antag^*| ≥ 0.02
L229, I272, F273, Q277, Y311, L321, V324, M325, M330, I382, Q401, E462, **Y464**	L229, Q277, **S280**, L309, Y311, V313, F318, L321, M325, I354, M355, F378, K448, E451, E462

Bold residues represent the key residues identified from previous studies.

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
