# Peer review of "A Molecular Dynamics Approach to Explore the Intramolecular Signal Transduction of PPAR-α"

_ijms, 2019, doi:10.3390/ijms20071666_

Reviewer 1 Report

The paper describes  the exploration of signal transduction in PPAR-alpha with molecular dynamics simulations and essentiasl dynamics analysis. The subject is challenging and the authors succeed in outlining the difference in the structure and dynamics od apo, agonist and antagonist bound protein. Partcularly interesting is the use of essential dynamics and network centrality analysis. The last one in particular confers originality to the paper. Only a couple of suggestions to improve the paper:

1)Pag. 5 line 166 The authors refer to contour maps showing " two minimal energy clusters and large structural distribution (Figure 4A and Figure 5D,F)" Actually only one minimum is evident to me in figure 5D, while two are evident in figure 4A. Where is the second minimum in 5D? Minima should be highlighted, maybe with circles.

2)Pag. 6 line 194 "...to the RMSD of backbone atoms..." It is not clear the reference structure(s) with respect to which RMSD is calculated.

3)Pag 9 lines 245 and 240. In line 245 residues are selected according to |CBApo -CBAgo| >0.02 while in line 248 it is said that they are plotted according to |CBApo -CBAgo| >0.03 (and the same for apo and antagonist). Is it a typo or there is a justification fot that? It should de clarified.

On the whole I reccomend acceptation of the manuscript with minor revisions.

Author Response

Reviewer #1

 Comments and Suggestions for Authors

 The paper describes the exploration of signal transduction in PPAR-alpha with molecular dynamics simulations and essentiasl dynamics analysis. The subject is challenging and the authors succeed in outlining the difference in the structure and dynamics od apo, agonist and antagonist bound protein. Partcularly interesting is the use of essential dynamics and network centrality analysis. The last one in particular confers originality to the paper. Only a couple of suggestions to improve the paper:

 1)Pag. 5 line 166 The authors refer to contour maps showing " two minimal energy clusters and large structural distribution (Figure 4A and Figure 5D,F)" Actually only one minimum is evident to me in figure 5D, while two are evident in figure 4A. Where is the second minimum in 5D? Minima should be highlighted, maybe with circles.

à Thank you for your suggestion. As per your comment, we have highlighted the minimal energy clusters using black dotted circles. Please check revised figures 4 and 5.

 2)Pag. 6 line 194 "...to the RMSD of backbone atoms..." It is not clear the reference structure(s) with respect to which RMSD is calculated.

à In line with the reviewer’s comment, we have re-phrased the sentence in the revised manuscript as follows:

“The lowest conformational energy structures for apo, agonist- and antagonist-bound forms were obtained by plotting the total conformational energy of the system relative to the RMSD of the protein backbone atoms (calculated with respect to the equilibrated structure), which are highlighted as red circles in Figure 6C.”

 3)Pag 9 lines 245 and 240. In line 245 residues are selected according to |CBApo -CBAgo| >0.02 while in line 248 it is said that they are plotted according to |CBApo -CBAgo| >0.03 (and the same for apo and antagonist). Is it a typo or there is a justification fot that? It should de clarified.

à In Table 2, we accounted the key residues satisfying the conditions |CBApo -CBAgo| >0.02 and |CBApo -CBAntag| >0.02. Whereas, in the figure 8, when we mapped the residues that satisfy the above condition, it appears crowded due to the presence of several residues. Hence, for visualization feasibility, we showed only the residues satisfying the conditions |CBApo -CBAgo| >0.03 and |CBApo -CBAntag| >0.03.

 On the whole I reccomend acceptation of the manuscript with minor revisions.

à We thank the reviewer for positive comments and acceptance of our manuscript.

Reviewer 2 Report

The Manuscript by Batish Shaherin et al reports a predictive analysis of PPAR structure.

 In my opinion, this paper is well done in all its part: scientific point of view, how is written, and the meaning of this paper. Exploring the intramolecular signal transduction of PPAR-a, with MD approach,  could open to new study about medical chemistry.

 I accept this paper and wait for the next one to understand the effective PPAR-a modulators.

 Author Response

Reviewer #2

 Comments and Suggestions for Authors

 The Manuscript by Batish Shaherin et al reports a predictive analysis of PPAR structure. In my opinion, this paper is well done in all its part: scientific point of view, how is written, and the meaning of this paper. Exploring the intramolecular signal transduction of PPAR-a, with MD approach,  could open to new study about medical chemistry. I accept this paper and wait for the next one to understand the effective PPAR-a modulators.

à We thank the reviewer for his/her encouraging view and acceptance of our manuscript.